# LESS LiDAR: A Full-Waveform and Discrete-Return Multispectral LiDAR Simulator Based on Ray Tracing Algorithm

Yaotao Luo, Donghui Xie *, Jianbo Qi, Kun Zhou, Guangjian Yan and Xihan Mu

State Key Laboratory of Remote Sensing Science, Faculty of Geographical Science, Beijing Normal University, Beijing 100875, China; it@mail.bnu.edu.cn (Y.L.); jianboqi@bnu.edu.cn (J.Q.); kunzhou@mail.bnu.edu.cn (K.Z.); gjyan@bnu.edu.cn (G.Y.); muxihan@bnu.edu.cn (X.M.)

* Correspondence: xiedonghui@bnu.edu.cn

**Abstract:** Light detection and ranging (LiDAR) is a widely used technology for the acquisition of three-dimensional (3D) information about a wide variety of physical objects and environments. However, before conducting a campaign, a test is typically conducted to assess the potential of the utilized algorithm for information retrieval. It might not be a real campaign but rather a simulation to save time and costs. Here, a multi-platform LiDAR simulation model considering the location, direction, and wavelength of each emitted laser pulse was developed based on the large-scale remote sensing (RS) data and image simulation framework (LESS) model, which is a 3D radiative transfer model for simulating passive optical remote sensing signals using the ray tracing algorithm. The LESS LiDAR simulator took footprint size, returned energy, multiple scattering, and multispectrum LiDAR into account. The waveform and point similarity were assessed with the LiDAR module of the discrete anisotropic radiative transfer (DART) model. Abstract and realistic scenes were designed to assess the simulated LiDAR waveforms and point clouds. A waveform comparison in the abstract scene with the DART LiDAR module showed that the relative error was lower than 1%. In the realistic scene, airborne and terrestrial laser scanning were simulated by LESS and DART LiDAR modules. Their coefficients of determination ranged from 0.9108 to 0.9984. Their mean was 0.9698. The number of discrete returns fitted well and the coefficient of determination was 0.9986. A terrestrial point cloud comparison in the realistic scene showed that the coefficient of determination between the two sets of data could reach 0.9849. The performance of the LESS LiDAR simulator was also compared with the DART LiDAR module and HELIOS++. The results showed that the LESS LiDAR simulator is over three times faster than the DART LiDAR module and HELIOS++ when simulating terrestrial point clouds in a realistic scene. The proposed LiDAR simulator offers two modes for simulating point clouds: single-ray and multi-ray modes. The findings demonstrate that utilizing a single-ray simulation approach can significantly reduce the simulation time, by over 28 times, without substantially affecting the overall point number or ground pointswhen compared to employing multiple rays for simulations. This new LESS model integrating a LiDAR simulator has great potential in terms of simultaneously simulating LiDAR data and optical images based on the same 3D scene and parameters. As a proof of concept, the normalized difference vegetation index (NDVI) results from multispectral images and the vertical profiles from multispectral LiDAR waveforms were simulated and analyzed. The results showed that the proposed LESS LiDAR simulator can fulfill its design goals.

**Keywords:** radiative transfer; ray tracing; LiDAR; LESS; DART



## 1. Introduction

As an active detection technique, light detection and ranging (LiDAR) has incomparable advantages over passive remote sensing technology in terms of obtaining structural information about three-dimensional (3D) objects. It is increasingly used in many fields,

such as building modeling [1], topography acquisition [2], and vegetation architecture retrieval [3].

Traditional LiDAR systems can quickly collect high-accuracy 3D data at a designated wavelength. Passive multi- or hyperspectral imagers can collect rich spectral information about ground objects, but they lack 3D spatial information. Therefore, active LiDAR and passive optical (multi- or hyperspectral) remote sensing images are often integrated to obtain both spatial and spectral information about objects, for example, by estimating the 3D chlorophyll content distributions of trees [4]. Therefore, models that can simulate both passive and active optical remote sensing signals simultaneously need to be developed to facilitate the application of multi-source remote sensing data fusion, which is a research gap from previous studies since there are only a small number of models mentioned in the following paragraphs that can simulate both passive and active optical remote sensing signals simultaneously.

Over the past several decades, to explain the characteristics of remote sensing signals and understand the interaction between light and vegetation, some vegetation radiative transfer models have been developed to simulate passive optical remote sensing signals [5–7]. With the development of computer technology, 3D radiative transfer models have become increasingly sophisticated by taking more details of canopies and environmental impact factors into account, with models such as the discrete anisotropic radiative transfer (DART) [8,9], FLIGHT [10], RAYTRAN [11], PS and SPS [12], and Rayspread [13]. Three-dimensional radiative transfer models have the advantages of higher simulation accuracy and fewer hypotheses and simplifications, however, their weaknesses have hampered their application, including the labor-intensive process to construct a large-scale scene taking many details of canopies and environmental impact factors into account, and the time-consuming calculations. The large-scale remote sensing data and image simulation framework (LESS) takes advantage of computer technologies to quickly simulate remote sensing images based on 3D large-scale scenes with fewer structural simplifications, and its accuracy is evaluated by comparison with other typical models as well as field measurements [14]. Note that LESS simulates RS images quickly, but that constructing elaborate scenes quickly is out of the scope of this article.

LiDAR simulation models were developed later than passive radiative transfer models. Most LiDAR models were developed based on ray tracing algorithms, such as LITE [15], extended FLIGHT [16], DART LiDAR module [17,18], and HELIOS [19]. However, computational efficiency and ease of use are still issues that hamper the application of these models. Compared to its predecessor HELIOS, HELIOS++ had a greatly improved ease of configuration and reduced runtime [20], while a friendly GUI (graphical user interface) and some prepared 3D structural scenes are still expected to help its users. Although lacking a GUI, HELIOS++ offers integration with popular software such as CloudCompare, QGIS, and Blender in order to facilitate the generation of input data.

Those separate active or passive radiative transfer models cannot fulfill the requirement of multi-source data fusion when we are facing so rich and varied remote sensing data. Some LiDAR models have been extended based on passive optical radiative transfer models, such as FLIGHT [16] and DART [17,18], which can simulate optical remote sensing images and LiDAR signals. This kind of combined model is significant because it enables investigators to quantitatively evaluate the accuracy of the fusion of LiDAR and optical image data and explore the applications of fused passive and active data.

In recent years, the development of multi- or hyperspectral LiDAR instrument has become increasingly popular [21], which can combine the characteristics of 3D structural and spectral information and provide more accurate remote sensing detection and physiological and biochemical vegetation parameter retrievals than other approaches. For example, Rall and Knox demonstrated a vegetation index LiDAR technique using two laser diodes straddling the chlorophyll absorption peak at 680 nm [22]. Pan et al. applied a multispectral LiDAR point cloud data with three channels to classify land cover [23]. Hakala et al. presented the concept and the first prototype of a full-waveform hyperspectral terrestrial laser

scanner, which makes it possible to study efficiently, e.g., the 3D distribution of chlorophyll or the water concentration in vegetation [24]. Although most multi- or hyperspectral LiDAR instruments are still at the prototype or developmental stage, a multi- or hyperspectral LiDAR model is required to optimize the design and explore its applications.

Our objective is to develop a LiDAR simulator based on the LESS model, capable of accurately simulating both single-wavelength and multispectral LiDAR as well as passive optical remote sensing images. Unlike existing simulators that rely on simplified assumptions, our simulator employs a ray tracing algorithm that incorporates the complex characteristics of real-world scenes. This approach ensures a more realistic representation of LiDAR instruments across different platforms, taking into account factors such as the location, direction, and wavelength of each laser pulse emitted.

An important aspect of our proposed LiDAR simulator is its utilization of the original LESS model's advantageous features. These include a user-friendly graphical user interface (GUI), allowing for easy configuration and generation of visually compelling 3D scenes. By harnessing these capabilities, we are able to provide a proof-of-concept demonstration through the simulation of multispectral images alongside their corresponding multispectral LiDAR waveforms.

In order to assess the accuracy of our proposed LiDAR simulator, we conducted evaluations focusing on waveforms and 3D point outputs using the DART LiDAR module. Additionally, we compared the performance of our LESS LiDAR simulator with that of both the DART LiDAR module and HELIOS++. By considering both accuracy and performance results together, it becomes apparent that our study introduces novelty in two key aspects: first, LESS enables efficient simulations without sacrificing accuracy; second, its reduced simulation time renders it a practical choice for researchers in this field.

Our proposed LiDAR simulator offers multi-ray and single-ray modes for simulating point clouds which are also offered by models like HELIOS++. The first mode employs multiple ray casting queries to simulate the divergence of a laser beam; hence it is referred to as the "multi-ray point cloud simulation". This mode generates simulated full waveforms containing valuable additional information that can offer profound insights into scanned objects. The second mode, known as the "single-ray point cloud simulation mode", treats the beam as a single ray. This mode also presents itself as a valuable choice for scenarios where efficiency is prioritized, especially in situations involving shorter distances and smaller laser footprints.

## 2. Data

### 2.1. Abstract Scenes

Several test cases were designed and conducted to simulate LiDAR waveforms. Lambertian scene objects were built, including pads, steps, slopes, and plant crowns (Table 1 and Figure 1), with different reflectances and LiDAR device altitudes. Note that the heterogeneous cylinder is from the radiative transfer model intercomparison (RAMI, https://rami-benchmark.jrc.ec.europa.eu/, accessed on 10 September 2023). The results were compared against references generated by the DART LiDAR module [17,18].

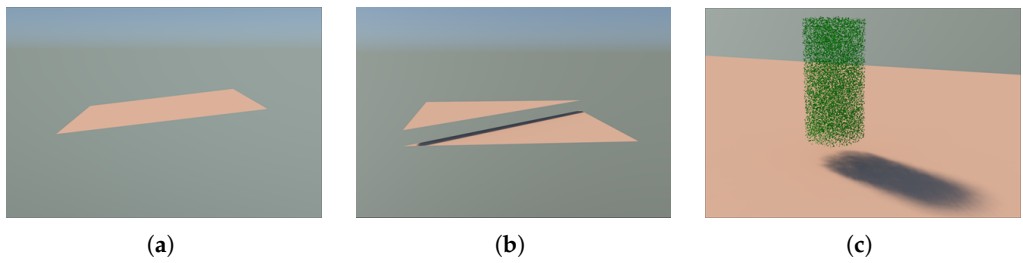

(a)        (b)        (c)

**Figure 1.** Illustrations of test scenes. (**a**) Pad/slope (by rotation). (**b**) Small steps/big steps (changing height difference). (**c**) Heterogeneous cylinder and pad.

**Table 1.** Descriptions of the objects in the test scenes.

| Object | Description |
| --- | --- |
| Pad | Horizontally placed square 2 m above the ground with a side length of 40 m |
| Slope | An inclined pad with a rise of 3 units for every 50 units of run and a center that is 2 m above the ground |
| Small steps | Steps with a height difference of 0.9 m. The lowest step is 0.5 m above the ground |
| Big steps | Steps with a height difference of 2 m. The lowest step is 0.5 m above the ground |
| Heterogeneous cylinder | A cylindrical object placed upright with a height of 12 m and a radius of 3 m that contains 17,999 discs of negligible thickness with a radius of 0.05 m, the bottom-most of which is 2 m above the ground |

*2.2. Realistic Scene*

Some tree models and scenes from the radiative transfer model intercomparison (RAMI, https://rami-benchmark.jrc.ec.europa.eu/, accessed on 10 September 2023) [25] are often taken as the standard structure to evaluate new radiative transfer models. In this study, the Järvselja Birch stand (summer) scene, including the tree models (Figure 2) and their spectral characteristics (Figure 3) were used to construct an example scene. The scene was generated on the basis of 12 individual tree representations. These were distributed among the following tree species: ACPL = maple (*Acer platenoides*), BEPE1-4 = birch (*Betula pendula*), ALGL2 = alder (*Almus glutinosa*), TICO2-5 = linden (*Tilio cordata*), POTR1 = poplar (*Populus tremuloides*), and FREX = ash (*Fraxinus exelsior*). An overview of the characteristics of these tree representations can be found on the RAMI webpage (https://rami-benchmark.jrc.ec.europa.eu/, accessed on 10 September 2023). The scene was a subset of the Järvselja Birch stand (summer) scene, and the selected tree models were mainly *Betula pendula* and *Tilio cordata* models. The tree heights of *Betula pendula* vary from 19.86 m to 30.51 m, and those of *Tilio cordata* vary from 11.27 m to 20.70 m. Thus, there were several *Tilio cordata* crown models beneath the crown models of *Betula pendula*.

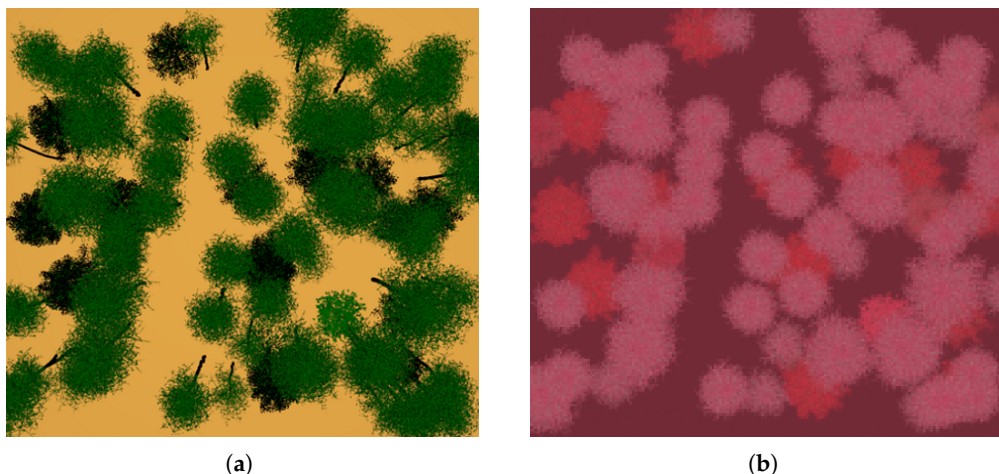

(**a**)  (**b**)

**Figure 2.** Illustration of the example scene generated using the data from the radiative transfer model intercomparison (RAMI). It can be seen that some tree crowns are beneath others in the image. (**a**) The example scene generated using data from RAMI is shown in the the large-scale remote sensing data and image simulation framework (LESS) three-dimensional (3D) viewer. Colors are used to distinguish different elements, not to reflect the radiometric properties. (**b**) An orthographic false color image generated by LESS. The near infrared band is shown as red. Red bands are shown in green. Green bands are shown in blue.

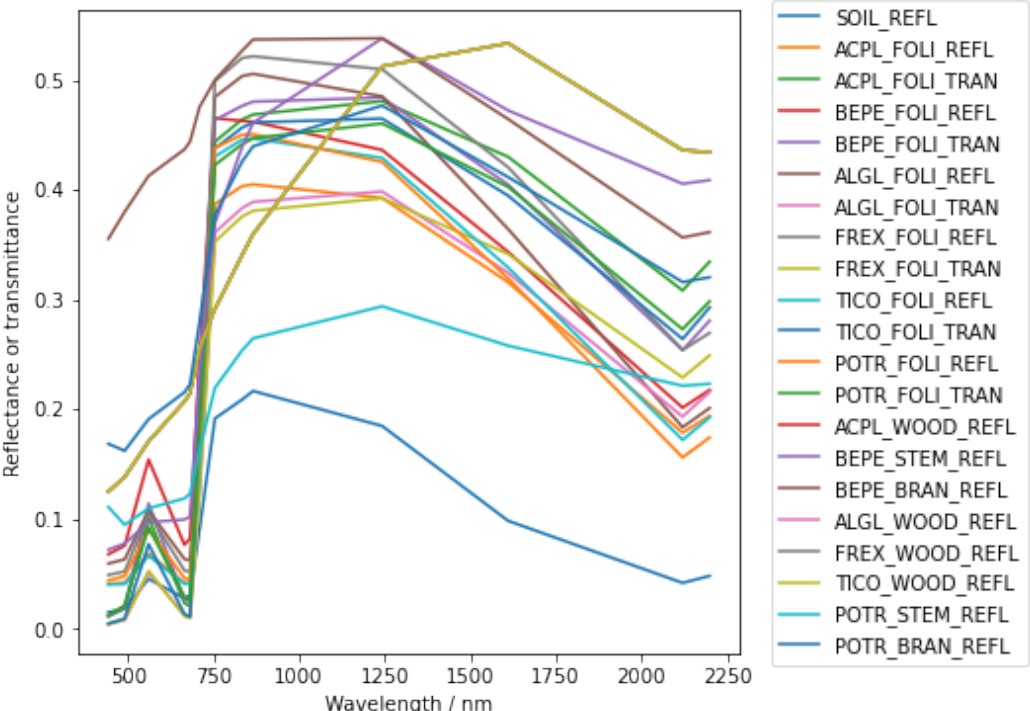

**Figure 3.** Optical characteristics of the scene elements.

## 3. Method

LESS is one of the 3D optical radiative transfer models and it ranges from visible light to thermal infrared waves [14]. Forward photon tracing and backward path tracing are implemented in LESS. LESS has been evaluated in comparison with other models with respect to accuracy, as well as field measurements in terms of directional reflectance and pixel-wise simulated image comparisons, which show very good agreement.

LESS supports the simulation of large-scale spectral images based on 3D landscapes and provides an easy-to-use GUI and related tools to construct 3D scenes. A LiDAR simulator is developed based on forward photon tracing to extend the functionality of LESS so that it can simulate active LiDAR data based on the same scenes used for passive optical images as well.

### 3.1. Principles of LiDAR

LiDAR is a direct extension of conventional radar (radio detection and ranging) techniques to very short wavelengths [26]. Its main purpose is to range and detect objects.

The principle of using LiDAR to scan objects can be simply described as follows. A laser emits pulses (the time-dependent function of which is denoted $W_t$) towards the object (whose properties that are concerned can be accounted for by introducing the target profile $P_{lidar}$) and then some of the energy is scattered back to the sensor where it is measured with an optical receiver (which has a system response function denoted $\Gamma$). A timer measures the travel time of the pulse from the laser to the object and back to the laser scanner. The distance of the sensor to the object can be calculated based on the time of flight. Symbolically, this is an equation [27,28] representing the fundamental relations between the characteristics of the sensor, the target, and the received signal:

$$\tilde{P}_{lidar} = W_t * P_{lidar} * \Gamma, \tag{1}$$

where $*$ is the convolution operator. That is, the shape of the LiDAR waveform ($\tilde{P}_{lidar}$) is a convolution of the time-dependent function of the emitted pulse ($W_t$) with the target profile ($P_{lidar}$) and the system response function ($\Gamma$) [26].

For spatially distributed targets the return signal is the superposition of echoes from scatterers at different ranges/times, and the echo pulse is exactly the result of a convolution of those terms [27]. In this study, the ideal situation is considered. Therefore, the system response function is a Dirac delta function.

Hence, waveform simulations were performed in two steps. First, the time (or distance)-dependent profile of the target was modeled (Sections 3.2.1 and 3.2.2). Secondly, the profile was convolved with the emitted pulse to obtain the waveform (Section 3.2.3).

Generally, there are two categories of LiDAR data formats, including point cloud (discrete returns) and full waveform. These data are offered by laser scanners on different platforms considering the location, direction, and wavelength of each emitted laser pulse. In this study, a unified algorithm is used to simulate full waveform firstly, and then point cloud data are calculated based on Gaussian decomposition of the full waveform (Section 3.2.4).

### 3.2. LiDAR Simulation Based on Ray Tracing Algorithm

In this section, the ray tracing algorithm is applied to simulate a LiDAR signal. According to the ray tracing algorithm, a virtual laser is supposed to emit virtual laser beams in 3D space. The returns are obtained by calculating the intersection points of the virtual laser beams and the scene elements. If an intersection point can be "seen" in the field of view of the sensor, scattered energy is received by the receiver (Figure 4).

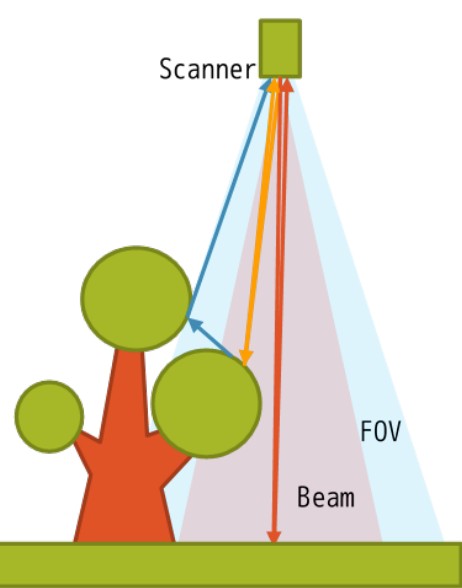

**Figure 4.** Illustration of the light detection and ranging (LiDAR) simulation procedure. A virtual laser is supposed to emit virtual laser beams in 3D space. To simulate the divergence of a laser beam, multiple ray casting queries are used to approximate a cone (not all sample rays but rather the representative rays are shown). If an intersection point can be "seen" in the field of view of the sensor, scattered energy is received by the receiver.

#### 3.2.1. Generating a LiDAR Pulse

Laser has the advantage of good collimation with limited divergence angle. Generally, a laser beam can be regarded as a cone with a very small apex angle. The longer the beam path is, the larger the footprint of the laser beam is. To simulate the divergence of a laser beam, multiple ray casting queries are used to approximate the cone.

To discretize rays within one laser beam, some points are sampled in the circular cross-section the cone possesses. The origin of the rays is the apex of the cone. The rays are sent towards the sample points. The point inside the circular cross-section is also a vertex of a square grid within the cross-section. Let the radius of the circle be 1. Then, the side

length of the grid cell is $2/N_s$, where $N_s$ determines the sampling density and can be set by the user (Figure 5). The larger the number of samples in one beam is, the more accurate the simulated signal is, and the slower the calculation procedure is.

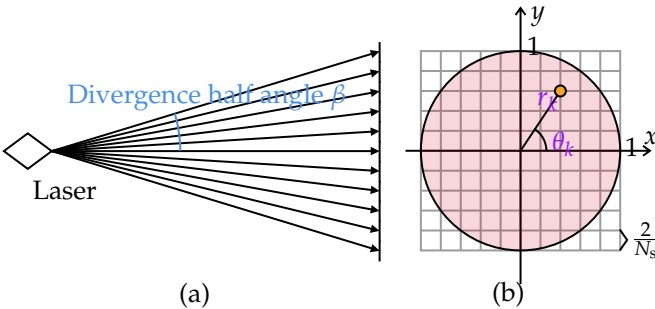

**Figure 5.** Illustration of the cone of a laser beam. (**a**) Side view of the beam. (**b**) Beam footprint.

Given all the sample points inside the circle whose polar coordinates are $(r_k, \theta_k)$, $k = 1, 2, ..., N$, where $N$ is the number of samples ($N \approx \pi N_s^2/4$ here) and $r_k$ and $\theta_k$ are the distance and angle of the $k$th sample, the weight of the initial ray (subscript 0) assigned to the emitted ray corresponding to the sample $(r_k, \theta_k)$ is

$$P_{0,k} = P \frac{W_r(r_k)}{\sum_{k=1}^{N} W_r(r_k)} \tag{2}$$

where $P$ is the pulse energy and $W_r(r_k)$ is a non-increasing function, because the energy reduces from the center to the edge of the footprint. Therefore, a Gaussian function $W_r(r_k) = \exp\left(-\frac{r_k^2}{2\sigma_r^2}\right)$ is used to depict the spatial distribution of energy across the beam path, where $\sigma_r$ is the deviation of the spatial energy distribution.

3.2.2. Pulse Propagation

After rays are sent from the light source, the pulse propagation is traced. This section presents the calculation method for the intersections between rays and objects and the simulation of scattering rays with directions and weights.

a. Intersections

To handle the scenes and calculate the intersections, Mitsuba [29], a research-oriented rendering system, is used in this study. In fact, the implementation of LESS is based on the open source ray tracing code Mitsuba. Mitsuba provides a flexible plugin architecture, which makes it possible for developers to extend LESS with new functionalities without knowing and recompiling the whole system [14].

For each intersection, two new rays are generated. A ray is sent directly towards the LiDAR sensor with a weight $P_{\text{lidar},n+1,k}$. The other ray is sent to a randomly selected direction with a weight $P_{n+1,k}$, where $n$ represents the number of scattering orders for the $k$th ray. The calculations of the direction and the weight of the two rays are described as follows.

b. The ray sent towards the LiDAR sensor

The ray sent directly towards the LiDAR sensor has a weight

$$P_{\text{lidar},n+1,k} = \chi \frac{\rho \cos\theta_l}{\pi} P_{n,k} \Omega, (n = 0, 1, \dots), \tag{3}$$

where $\chi$ has value of 1 if the intersection point is in the field of view (FOV) and visible to the sensor, and 0 otherwise, $\rho$ is the reflectance of the surface at the intersection point, and $\theta_l$ is the view zenith angle in relation to the surface normal. Note that the scene elements are

assumed to be Lambertian with known reflectance in this paper. The solid angle can be given as

$$\Omega = \frac{A_{\mathrm{t}} \cos \theta_{\mathrm{r}}}{S^2},$$ (4)

where $A_{\mathrm{t}}$ is the aperture area of the telescope, $S$ is the distance between the intersection point and the sensor, and $\theta_{\mathrm{r}}$ is the angle between the viewing direction of the sensor and the opposite direction of the ray from the intersection point to the sensor.

If scattered energy is received by the telescope of the receiver, an incident angle should be smaller than half the dispersion angle of the FOV. Let $\beta_{\mathrm{FOV}}$, $r_t$, $\vec{l}$, and $V_1$ denote half the dispersion of the FOV, the telescope aperture radius, the unit vector of device orientation, and the vector representing the position of the device, respectively. The vector $V_{\mathrm{c}}$ representing the reception convergence point is

$$V_{\mathrm{c}} = V_1 - \frac{r_{\mathrm{t}}}{\tan \beta_{\mathrm{FOV}}} \vec{l}.$$ (5)

Let $V_n$ denote the intersection point. The incident angle is calculated as

$$\theta_{\mathrm{r}} = \arccos\left(-\frac{\overrightarrow{V_{\mathrm{c}}V_n}}{|\overrightarrow{V_{\mathrm{c}}V_n}|} \cdot \vec{l}\right).$$ (6)

If $\theta_{\mathrm{r}} < \beta_{\mathrm{FOV}}$, the telescope can receive scattered energy (Figure 6).

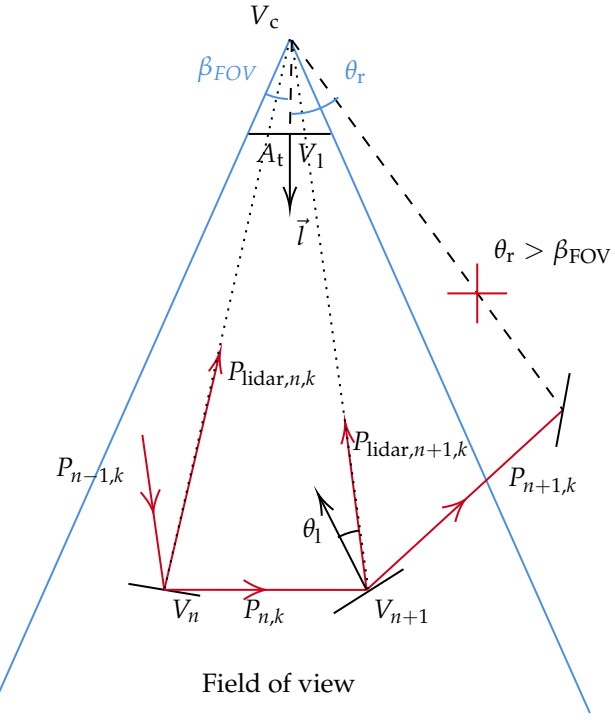

**Figure 6.** Illustration of pulse propagation scheme. If a ray $k$ intersects a scene element $V_n$, a new ray is cast towards the receiver firstly, and the contribution is calculated. Then, another ray is generated to sample the next scattering order. Here, the ray escapes the field of view after the interaction at $V_{n+1}$. Note that in reality, the receiver and the emitter are at the same location in the LiDAR sensor.

c. The ray sent to a randomly selected direction

For each scattering event, the $k$th ray is pointed in a random direction with a weight

$$P_{n+1,k} = (1 - a)P_{n,k}$$ (7)

where $P_{n,k}$ is the weight of the incident ray, $n$ denotes the scatter order, where $a$ is the absorption, namely, $a = 1 - \rho - \tau$, where $\rho$ is the reflectance, and $\tau$ is the transmittance of the scene element. Then, $(1 - a)$ represents the proportion that is not absorbed in this scattering event.

The scattering direction is also supposed to be decided. First, a decision is made equally and randomly between a reflection or a transmission. Secondly, the direction of the ray is a cosine-weighted vector $(x, y, z)$ sampled on the unit hemisphere.

To sample a cosine-weighted vector on the unit hemisphere, a random point on a unit disk is generated. Then, the point is projected along the $z$-axis [30]. To generate a random point on the disk of unit radius, a random point $(u_1, u_2)$ on $[0, 1]^2$ (a square) is passed through a map between a square and a disk. The polar coordinates of the point on the disk are denoted as $(r, \varphi)$, where $r$ is the distance of the point from the disk center, and $\varphi$ is the angle from an arbitrary reference direction on the disk. The map is

$$(r, \varphi) = \begin{cases} (0, 0), & |r_1| + |r_2| = 0, \\ (r_1, (\pi/4)(r_2/r_1)), & r_1^2 > r_2^2, \\ (r_2, \pi/2 - (r_1/r_2)(\pi/4)), & r_1^2 \leq r_2^2, \end{cases} \tag{8}$$

where $r_1 = 2u_1 - 1$, and $r_2 = 2u_2 - 1$. Hence, the cosine-weighted vector on the unit hemisphere is $(x, y, z) = \left( r \cos \varphi, r \sin \varphi, \sqrt{1 - r^2 \cos^2 \varphi - r^2 \sin^2 \varphi} \right)$.

Then, a transformation is performed, which maps the direction vector from the local space to the world space.

Stop if the ray escapes the scene, or if the predefined number of scattering orders is reached.

### 3.2.3. Signal Recording

To obtain the waveform, the estimation of the target profile is convolved with the emitted pulse. The process is shown schematically in Figure 7. The weight $P_{\text{lidar},k}$ and the travel distance when the ray returns to the sensor are used. The received weights are binned into several bins according to their travel distances. The width of each bin is defined by the input acquisition rate $\Delta t$, which is related to the device specification. For example, the acquisition rate of Riegl LMS-Q680i is 1 ns. Then, the accumulation of these weights, denoted $P_{\text{lidar}}(t)$, is the estimation of the target profile.

The central part of a Gaussian function is used to fit the emitted pulse:

$$W_{\text{t}}(t) = \exp \left( -\frac{t^2}{2\sigma_{\text{time}}^2} \right), \quad -n_{\text{t}}\sigma_{\text{time}} \leq t \leq n_{\text{t}}\sigma_{\text{time}}, \tag{9}$$

where $\sigma_{\text{time}}$ is a positive real number standing for the standard deviation of the Gaussian function, and $n_{\text{t}}$ is the half pulse duration in number of sigma, then $n_{\text{t}}\sigma_{\text{time}}$ is the half pulse duration in nanoseconds. Here, the half pulse duration in number of sigma $n_{\text{t}}$ is one of the input parameters of the model. That is, in practice, the laser pulse has a curved shape and finite duration. In this study, we assume a Gaussian-shaped pulse defined by a half pulse duration at half peak ($t_{\text{half}}$ in nanosecond) and length of tails measured in standard deviations. It is known that the integral of the Gaussian distribution probability density function within 3 standard deviations of the mean (above or below) makes up 99.73% of the area under the curve. Therefore, $n_{\text{t}} \approx 3$ is sufficient to represent the pulse.

If the half pulse duration at half peak $t_{\text{half}}$ is given (for example, $t_{\text{half}} = 3.25$ ns is used to drive the airborne laser scanning (ALS) simulation here), the standard deviation $\sigma_{\text{time}}$ of the Gaussian function can be determined according to the equation

$$\frac{1}{2} = \exp \left( -\frac{t_{\text{half}}^2}{2\sigma_{\text{time}}^2} \right). \tag{10}$$

Solving for $\sigma_{\text{time}}$ gives

$$\sigma_{\text{time}} = \frac{t_{\text{half}}}{\sqrt{2 \ln 2}}. \tag{11}$$

Note, if the emitted pulse path is perpendicular to a perfectly planar surface, the received waveform will have one pulse with the same duration as the emitted pulse. This suggests a method to estimate the half pulse duration at half peak, $t_{\text{half}}$.

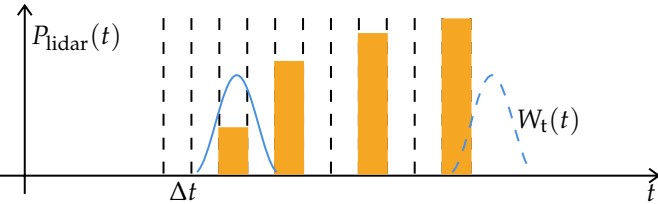

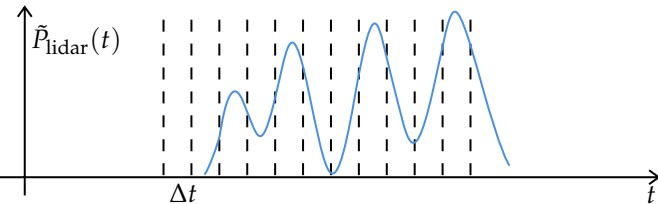

**Figure 7.** The process of the generation of the waveform. The received weights are binned into several bins according to their travel distances. The width of each bin is defined by the input acquisition rate $\Delta t$, which is related to the device specification. Then, the accumulation of these weights, denoted $P_{\text{lidar}}(t)$, is the estimation of the target profile. The waveform $\tilde{P}_{\text{lidar}}(t)$ is obtained by convolving $P_{\text{lidar}}(t)$ with the emitted pulse $W_{\text{t}}(t)$.

Once the estimation of the target profile, that is, the accumulation of the weights, $P_{\text{lidar}}(t)$, has been generated, and if the emitted pulse $W_{\text{t}}(t)$ has been given according to Equation (9), the waveform $\tilde{P}_{\text{lidar}}(t)$ is obtained by convolving $P_{\text{lidar}}(t)$ with the emitted pulse $W_{\text{t}}(t)$.

The profile and the pulse are represented as arrays in computer programming, specifically. Hence, denote the profile $P_{\text{lidar}}[i]$ and the emitted pulse $W_{\text{t}}[i]$, where $i$ is an integer. According to the discrete convolution formula [31], the waveform, denoted $\tilde{P}_{\text{lidar}}[i]$, is

$$\tilde{P}_{\text{lidar}}[i] = \frac{\sum_{k=-n}^{+n} P_{\text{lidar}}[i-k] W_{\text{t}}[k]}{\sum_{k=-n}^{+n} W_{\text{t}}[k]}, \tag{12}$$

where the integer $n$ controls the number of samples from $W_{\text{t}}(t)$, estimated by the equation

$$(2n+1)\Delta t = 2n_{\text{t}}\sigma_{\text{time}}. \tag{13}$$

### 3.2.4. Pulse Post-Processing

Gaussian decomposition enables the retrieval of point clouds [32]. The basic idea of Gaussian decomposition is to represent an approximate waveform $\hat{P}_{\text{lidar}}$ that is close to the original waveform $\tilde{P}_{\text{lidar}}$ as the sum of $M$ Gaussian functions:

$$\hat{P}_{\text{lidar}}(t) = \sum_{i=1}^{M} F_i \exp\left(-\frac{(t-t_i)^2}{2\sigma_{\text{p},i}^2}\right). \tag{14}$$

where $t_i$, $\sigma_{\text{p},i}$, and $F_i$ are the center, standard deviation, and amplitude of the $i$th Gaussian component, respectively.

When retrieving point clouds, $ct_i/2$ is the distance between a point and the sensor. Thus, the $i$th point from a waveform indexed $j$ can be presented as:

$$V_{i,j} = V_1 + \frac{1}{2}ct_i\vec{l} \tag{15}$$

where $\vec{l}$ is the unit vector of the pulse direction and $V_1$ is the vector representing the position of the device, which is also the pulse start position. The terrestrial laser scanning (TLS) tends to have an identical device position $V_1$ for each single-site scanning, while those LiDAR scanners mounted on moving platforms like ALS have a series of changing device positions.

### 3.3. LESS LiDAR Simulator

The LiDAR simulator is integrated into LESS [14], where easy-to-use graphical user interfaces and related tools for configuring the simulations are provided, such as constructing a complex 3D scene model in the wavefront object file type and setting the spectra of all components (background, tree branches, leaves, and so on). The OBJ file format is chosen as it is a commonly used format for 3D scenes. In case users have 3D data in other formats, conversion to OBJ format can be achieved using popular 3D software like Blender. As of now, LESS has the capability to accept OBJ files as input for 3D scenes.

The procedure introduced above simulates the LiDAR waveform and its points of one pulse. Multi-pulse data can be simulated by running the procedure in a loop. During multi-pulse acquisition, some parameters such as the sensor area and FOV half angle are constant terms, whereas the device position and the pulse direction are variables (Table 2). In waveform data processing, waveforms are decomposed into points, which are estimated locations of scene elements. Moreover, another simulation approach is also developed to generate the point cloud called a single-ray point cloud simulation, which is efficient for simulating point clouds of targets a short distance away and with a small divergence angle of the laser where the footprint is very small so the pulse is treated as one ray. The coordinates of the first intersection give the position of a point. The single-ray point cloud mode can obtain 3D information quickly and test the configuration parameters for the formal simulation.

**Table 2.** Parameters and their symbols and meanings in the LiDAR simulator.

| Parameter | Symbol | Meaning |
|---|---|---|
| Sensor area | $A_t$ | Aperture area of the telescope |
| Footprint half angle | $\beta$ | - |
| FOV half angle | $\beta_{FOV}$ | - |
| Pulse energy | $P$ | Energy of each pulse |
| Acquisition rate (period) | $\Delta t$ | Duration of each measurement. A 1 ns duration corresponds to a path of 30 cm, which, at the nadir and for scattering order 1, corresponds to a 15 cm altitude difference |
| Half duration (number of sigma) | $n_t$ | - |
| Half pulse duration at half peak | $t_{half}$ | This half pulse duration is used to compute the Gaussian pulse standard deviation (sigma) |
| Fraction at radius | - | Used to compute the standard deviation of the LiDAR energy Gaussian spatial distribution |
| Axial division | $N_s$ | The axial subcenter division of illumination |
| Max scattering order | - | If a ray reaches the maximum order of scattering + 1, it is considered lost |
| Device position (m) | $V_1(x, y, z)$ | For example, the terrestrial laser scanning (TLS) Position X, TLS Position Y, and TLS Position Z in TLS mode |
| Pulse direction (m) | $\vec{l}(x_d, y_d, z_d)$ | - |
| Minimum range (m) | - | Stored minimum waveform distance range from LiDAR |
| Maximum range (m) | - | Stored maximum waveform distance range from LiDAR |

In order to improve the processing speed, multi-threading processing is implemented. This uses the fact that each pulse is independently simulated with its own geometry.

The LESS LiDAR simulator supports the importation of actual and abstract configurations about the device and platform parameterization (e.g., platform path and angular/distance separation between pulses).

The actual configurations can be defined by an imported file setting the characteristics of each pulse (LiDAR position and orientation per pulse), which can be extracted from actual data. The LiDAR simulation based on the actual configuration is convenient to compare with actual data.

The abstract configurations can vary from platform to platform. This paper concentrates on terrestrial (TLS) and airborne (ALS) acquisitions. Two scanning modes for the platforms of TLS and ALS can be chosen. Both TLS and ALS modes simulate LiDAR signals based on the same procedure introduced above. The difference between TLS and ALS modes is the moving style of the platform, which can change the location and direction of each emitted laser pulse. The abstract configuration of the TLS mode is given by a regular grid with the following input parameters: (1) device position, and (2) grid mesh along the azimuth and zenith directions (angle range and angular separation between pulses). The abstract configuration of the ALS mode is given by a regular grid with the following input parameters: (1) start point and end point of the central axis of the swath region, and (2) grid mesh along the azimuth and range directions. Then, a waveform is simulated per grid node.

The waveform and the point coordinates are represented as arrays. Therefore, the waveform and point cloud data are column-oriented data stored in ASCII delimited text files compatible with libraries (for example, numpy or pandas in Python) and software (for example, Cloud Compare (www.cloudcompare.org, accessed on 10 September 2023)) that process data, which makes the information easy to access.

## 4. Results

The LESS LiDAR simulator is evaluated with the DART LiDAR module, including full waveforms and point clouds. The differences between two sets of data are assessed with the relative error, root mean square error (RMSE), or the coefficient of determination.

The relative error is defined by

$$\delta = \frac{y^* - y}{y^*} \times 100\% \tag{16}$$

where the reference value of the total received energy is $y^*$ and that of the output is $y$.

The formula of the the root mean square error is

$$RMSE = \sqrt{\frac{\sum_{i=1}^{n}(y_{t_i}^* - y_{t_i})^2}{n}} \tag{17}$$

where $y_{t_i}^*$ is the reference value at time $t_i$ and $y_{t_i}$ is the output at time $t_i$ ($1 \leq i \leq n$). The coefficient of determination $R^2$ is equal to the square of the correlation coefficient:

$$R^2 = \rho^2. \tag{18}$$

The correlation coefficient is defined as

$$\rho(\boldsymbol{y}, \boldsymbol{y}^*) = \frac{\mathrm{cov}(\boldsymbol{y}, \boldsymbol{y}^*)}{s \cdot s^*} \tag{19}$$

where $\mathrm{cov}(\boldsymbol{y}, \boldsymbol{y}^*)$ is the covariance of $\boldsymbol{y}$ and $\boldsymbol{y}^*$, and $s$ and $s^*$ are the standard deviation of $\boldsymbol{y}$ and $\boldsymbol{y}^*$, respectively.

*4.1. Waveforms*

4.1.1. Abstract Scenes

Waveforms are simulated based on the test scenes (Section 2.1) by the LESS and DART LiDAR modules. The parameters used in the LESS and DART LiDAR modules are the same. The configurations of the test cases (except for the heterogeneous cylinder) are shown in Table 3. The received energy values from the LESS and DART LiDAR simulations are compared for the simple and complex (heterogeneous cylinder) scenes. The relative errors for the simple test cases in Table 3 are 0.0%, which means there is no deviation between the two groups of simulated data. The relative error in the complex test case is about −0.205% for the full waveform (Table 4), which is very small. This proves that the LESS and DART LiDAR modules have a consistent simulation ability for these test cases. Additionally, we investigated the contribution of first scattering and multiple scattering effects. Previous studies [18] have emphasized the importance of multiple scattering in shaping waveforms within the near-infrared band. However, in our specific case, we observed a relative difference of only about 2% between these two effects. This implies that although multiple scattering can have some impact, it may not be highly significant in our scenario. Interested readers can refer to [18] for further information on the contribution of multiple scattering. By presenting Table 4, we aim to demonstrate that the LESS LiDAR simulator aligns well with the DART LiDAR module and provide a comprehensive understanding of the available options in LiDAR simulation software, particularly regarding first and multiple scattering effects.

**Table 3.** Configurations of the test cases and comparisons for the simple scenes between the LESS and the discrete anisotropic radiative transfer (DART).

| Index | Object | Reflectance | Altitude of Device/km | LESS Energy/ $10^{-13}$ J | DART Energy/ $10^{-13}$ J | Relative Error |
|---|---|---|---|---|---|---|
| 1 | Pad | 1.0 | 10 | 3.1844 | 3.1844 | 0 |
| 2 | Pad | 0.5 | 10 | 1.5922 | 1.5922 | 0 |
| 3 | Pad | 0.5 | 5 | 6.3713 | 6.3713 | 0 |
| 4 | Slope | 0.5 | 5 | 6.3686 | 6.3686 | 0 |
| 5 | Small steps | 0.5 | 5 | 6.3700 | 6.3700 | 0 |
| 6 | Big steps | 0.5 | 5 | 6.3599 | 6.3599 | 0 |
| 7 | Cylinder | 0.5 | 3 | - | - | - |
|  | Pad | 1.0 |  |  |  |  |

**Table 4.** Comparisons for the complex scene (heterogeneous cylinder).

|  | Energy/ $10^{-12}$ J | DART Energy/ $10^{-12}$ J | Relative Error | Root Mean Square Error (RMSE)/ $10^{-12}$ J |
|---|---|---|---|---|
| First-order scattering waveform | 1.8153 | 1.8154 | 0.006% | 0.0005 |
| Full waveform | 1.8618 | 1.8580 | −0.205% | 0.0005 |

The waveforms simulated by the LESS and DART LiDAR modules are directly compared in Figure 8. The retrieval distance range of a LiDAR signal is defined depending on the spatial region of interest. The horizontal axis represents the recorded time according to the ground return. It can be seen that all waveforms simulated by the LESS LiDAR simulator agree well with those by the DART LiDAR module.

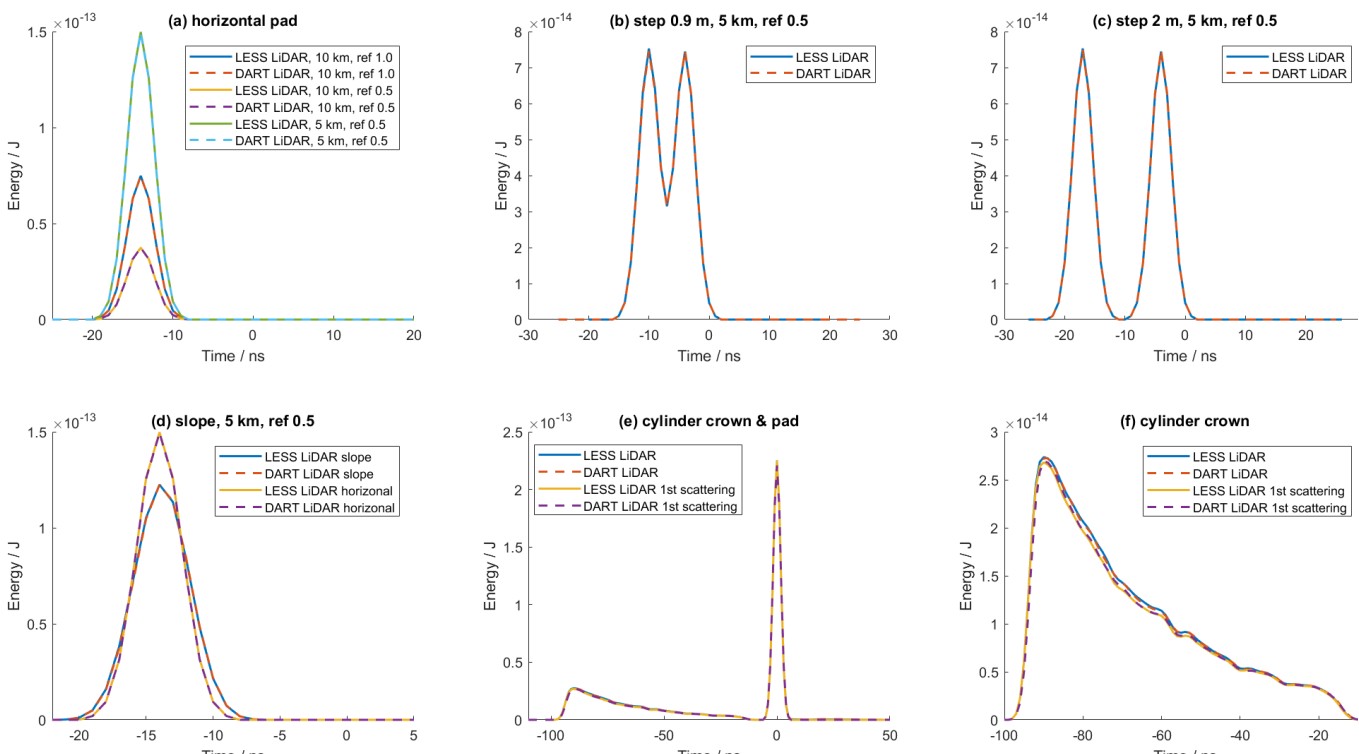

**Figure 8.** Comparison of the full waveforms simulated by LESS and DART LiDAR modules. (**a**) Waveforms of the pad with different parameters; (**b**) waveform of the small steps; (**c**) waveform of the large steps; (**d**) waveforms of the slope and horizonal pad; (**e**) waveforms of the cylinder and ground; (**f**) waveforms of the cylinder and ground (zoomed in).

The reflectance and distance impact the received energy (Figure 8a). The received energy is doubled if the reflectance is doubled. The received energy declines with the square of the range, which means that the received energy from distant targets is relatively small. The cases with the steps show multiple pulse returns (Figure 8b,c). The time differences between the peaks correspond to the height differences between the steps. The LiDAR acquisition rate is 1 ns. Thus, the absolute distance stored per bin is approximately equal to 15 cm. The time difference in the case of the small steps is 6 ns, corresponding to an estimated height difference of 0.9 m. That is, 1.95 m in the case of the large steps behaves in the same manner. The errors in the results are smaller than 15 cm. The broadening effect of the slope on the waveforms can be seen in Figure 8d. The case of the heterogeneous cylinder, representing a plant crown, is a complex scene, and Figure 8e demonstrates LiDAR acquisition with both multiple and single scattering orders. There are two peaks. The first one (left) is mainly caused by the cylinder of the tree, and the second (right) is the ground. The top of the cylinder is 14 m, corresponding to −94 ns (marked in Figure 8e). The first peak is zoomed in Figure 8f and some subtle differences can be seen between the waveform with multiple scattering orders, which is caused by random sampling of the simulation algorithms. Its RMSE is only $0.5 \times 10^{-15}$ J (Table 4).

### 4.1.2. Realistic Scene

The airborne waveforms are simulated based on the DART LiDAR module and the LESS LiDAR simulator to show the interactions between the laser energy and the complex forest canopy (Figure 2). The parameters of the LiDAR device (Table 5) and the scene input parameters (Figure 3) are used to drive the simulations.

**Table 5.** Configuration of airborne LiDAR device.

| Parameter/Unit | Value |
| --- | --- |
| Wavelength/nm | 1550 |
| Waveform sampling interval/ns | 1 |
| Laser beam divergence (half angle)/rad | 0.0012 |
| Altitude/m | 1000 |

There are 121 pairs of simulated waveforms. The RMSEs between the LESS and DART simulated waveforms range from $0.280 \times 10^{-14}$ J to $2.078 \times 10^{-14}$ J. The mean of the RMSEs is $0.866 \times 10^{-14}$ J. The coefficients of determination between the LESS and DART simulated waveforms range from 0.9108 to 0.9984. The mean of them is 0.9698. It can be seen that the waveforms simulated by the LESS LiDAR simulator agree well with those by the DART LiDAR module. Three are taken as examples to show the comparison between the simulated data (Figure 9). Some differences can be seen in Figure 9. These can be explained by the fact that if the distance between the sensor and some object is close to the edge of two adjacent bins, the contributions from the object may fall into different bins due to the different detailed implementations of LESS and DART. The paths and the trees are shown beside the waveforms in Figure 9, which are in accordance with the peak locations of the waveforms. Each wave peak corresponds to a horizontal level of the tree elements. It is obvious that waveforms can reflect the vertical structural characteristics of the canopy.

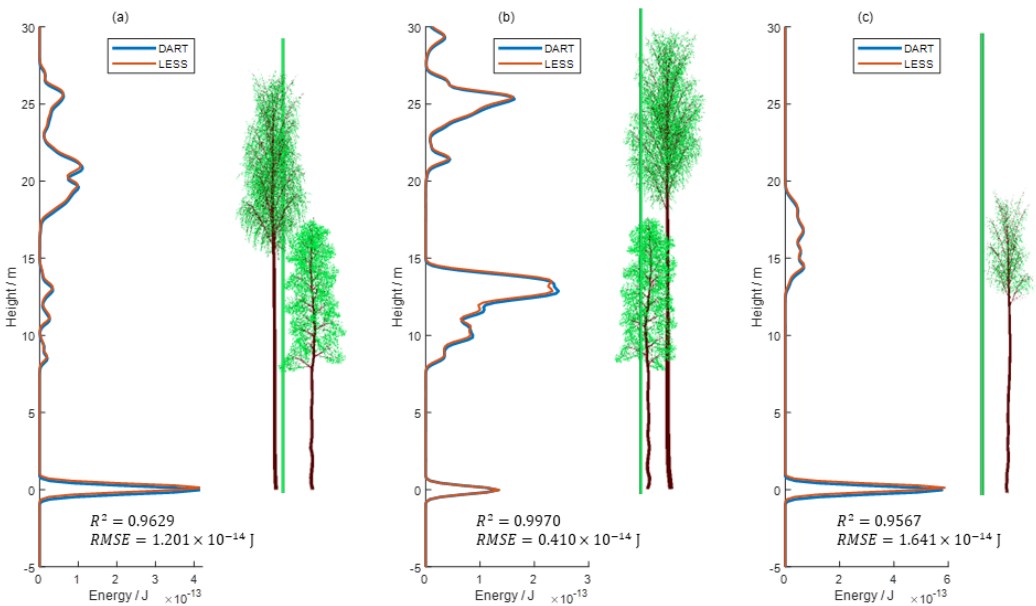

**Figure 9.** Comparisons of the simulated waveforms in the realistic scene (**a–c**) (Figure 2).

*4.2. Point Clouds*

4.2.1. Airborne Laser Scanning Point Cloud

Multiple returns can be obtained by decomposing the full waveforms [32]. There are 542 and 540 returns in the same scene (Figure 2) retrieved from the simulated data of the DART LiDAR module and the LESS LiDAR simulator, respectively. The procedures of retrieving returns in LESS and DART are not identical, therefore, the numbers of returns are slightly different. The numbers of returns with the layered height are quantitatively compared (Figure 10). The numbers of returns fit well and the coefficient of determination is 0.9986.

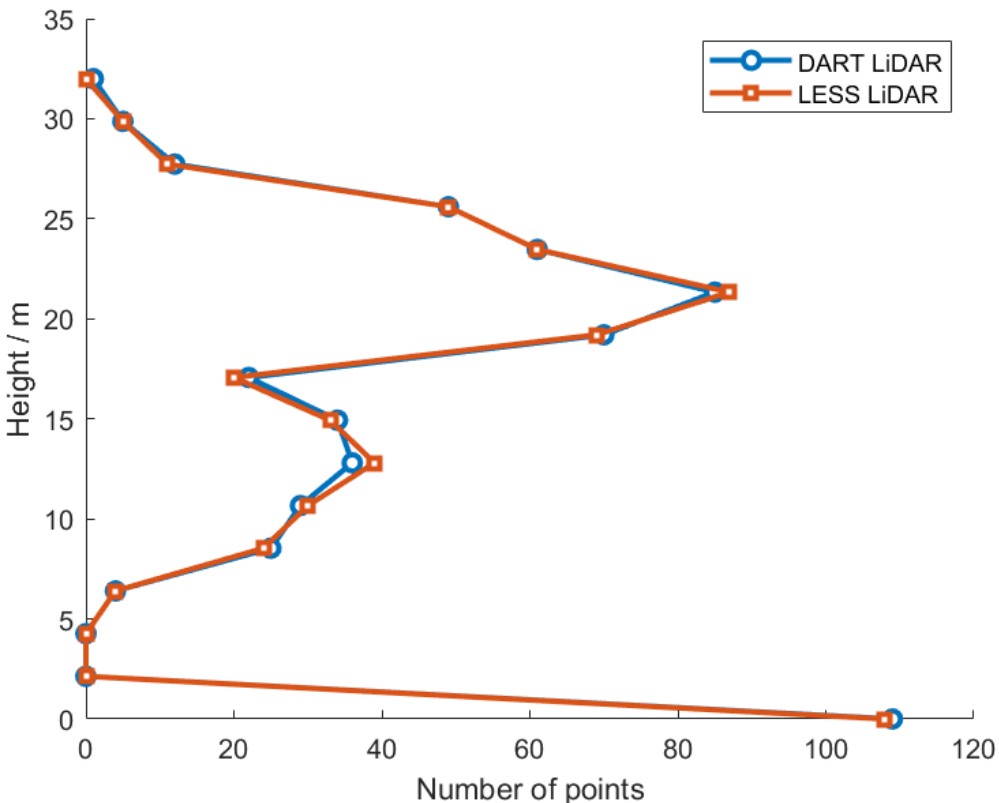

**Figure 10.** Comparison of point cloud data calculated based on the decomposition of full waveform. Comparison in terms of the layered number of points with the tree height in the realistic scene (Figure 2).

### 4.2.2. Terrestrial Laser Scanning Point Cloud

Terrestrial laser scanning (TLS) can obtain more dense point clouds with a smaller footprint of the laser than airborne laser scanning. TLS point clouds are simulated by the LESS LiDAR simulator and the DART LiDAR module using the same parameters. The scene input parameters (Figure 3) are used to drive the single-site point cloud simulation (Figure 11a). The point clouds are spatialized into layers with the height, horizontal cells, and 3D voxels. The number of points in these units are taken as the standard of the comparison between the LESS and DART simulated data. The layered comparison (Figure 11b) shows reasonable correspondence between the LESS and DART point clouds.

A horizontal comparison of the point number in cell units (5 m × 5 m) is shown in Figure 11d. It can be seen that the differences in most of the cells are close to 0 (blue), and several cells show overestimation (yellow) or underestimation (dark blue). There are several cells with high overestimation (light yellow) because of the small denominators in the relative difference. In fact, none of the absolute relative differences are larger than 1.

A 3D spatial comparison of the point number in voxels (cubes with 2 m sides) is shown in Figure 11e, which thoroughly represents the difference between the LESS and DART points. The number of LESS points within voxels agree well with the DART data. Most scatter points are distributed along the 1:1 line and the coefficient of determination between the two sets of data reaches 0.9849.

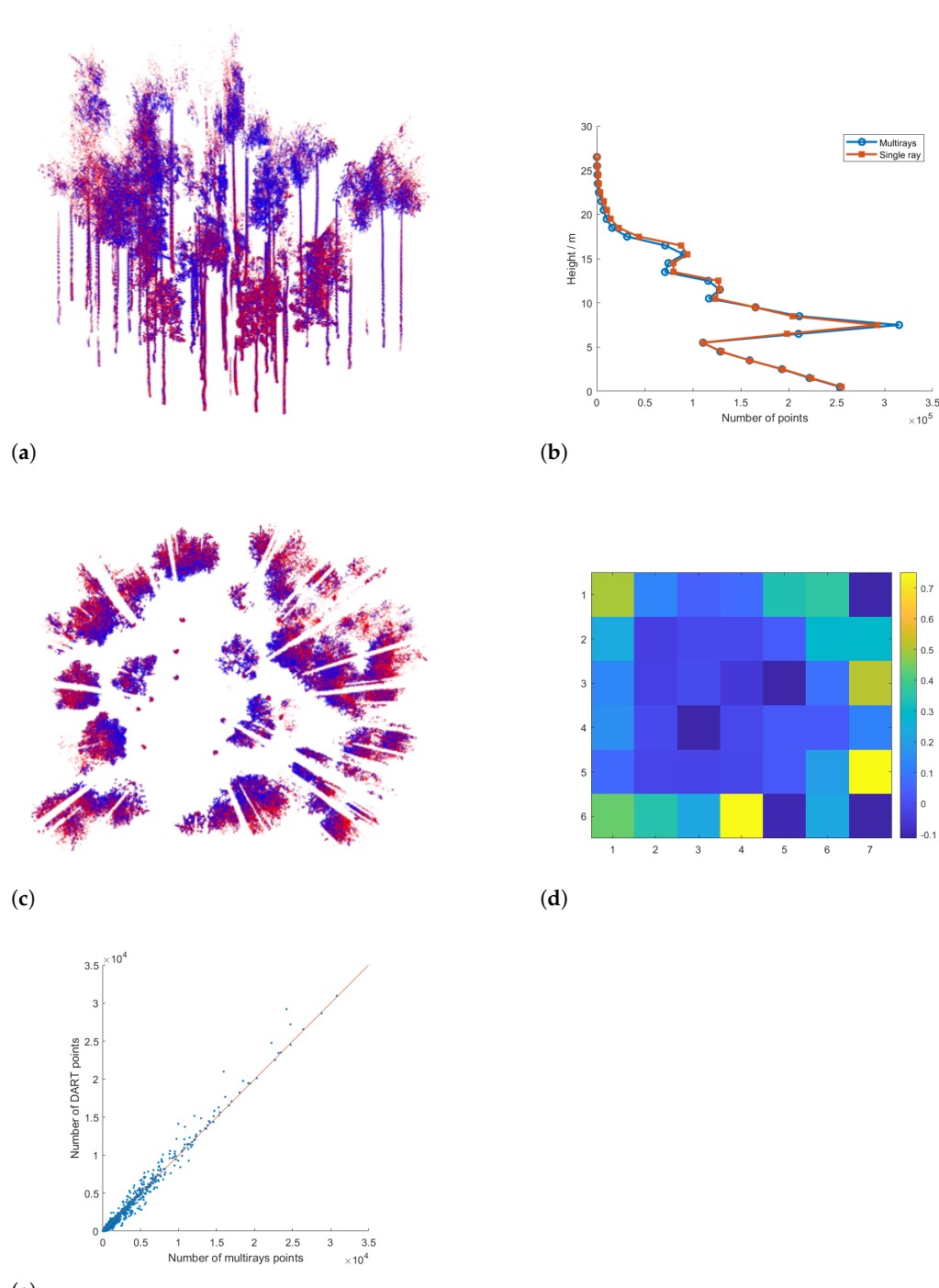

**Figure 11.** Comparison of LESS and DART point clouds. (**a**) DART (blue) and LESS (red) point clouds. (**b**) Comparison in terms of the layered number of points with the tree height. (**c**) The top view for the comparison of LESS and DART points. (**d**) The relative difference ((LESS-DART)/DART) of the number of points with horizontal cell size 5 m × 5 m. (**e**) Comparison of the number of LESS and DART points based on voxels (2 m × 2 m × 2 m).

## 5. Discussion

### 5.1. Multispectral LiDAR vs. Multispectral Imaging

Multispectral remote sensing images have been widely applied in many fields of Earth observation. Horizontal heterogeneity can be observed by multispectral imaging, but vertical variation may not be obtained directly. Fortunately, multispectral LiDAR

combines the characteristics of multispectral imaging and LiDAR techniques into a single instrument without any data registration. It provides more information than multispectral imaging or LiDAR alone.

Although some laboratories are developing multispectral LiDAR technology, proven multispectral LiDAR instruments for monitoring the vegetation are still lacking. To explore the ability of multispectral LiDAR, multispectral LiDAR signals are simulated by the LESS LiDAR simulator and compared with multispectral images in this section. The LESS model can simulate multispectral LiDAR and images based on the same set of parameters, including 3D scenes and the components' spectrum. An example using the normalized difference vegetation index (NDVI) [33] is designed to show that the proposed LiDAR model has the ability to simulate multispectral waveforms. Both the multispectral image and the waveforms are simulated based on the RAMI-V scene using the previously released LESS model and the proposed LESS LiDAR simulator, which is an extension of LESS. The LESS LiDAR simulator can simulate returned signals in multiple bands at the same time, instead of one band at a time, which can keep the simulated multispectral waveforms using the same ray paths and greatly save computing time.

The parameters in Tables 6 and 7 are used to drive the simulations of multispectral image and waveforms.

**Table 6.** Configuration of the multispectral image for computing the normalized difference vegetation index (NDVI) of the simulation.

| Parameter | Value |
|---|---|
| Sensor type | orthographic |
| Pixel size | 2 m × 2 m |
| View zenith angle/° | 0 |
| View azimuth angle/° | 180 |
| Solar zenith angle/° | 0, 30, 60 |
| Solar azimuth angle/° | 90 |

**Table 7.** Configuration of the simulated airborne laser scanner for computing the NDVI.

| Parameter | Value |
|---|---|
| Waveform sampling interval/ns | 1 |
| Laser beam divergence (half angle)/rad | 0.0012 |
| Altitude/m | 833 |

The values at wavelengths of 665 nm (a typical red band) and 754 nm (a typical NIR band) are chosen to compute the NDVI. The NDVI of the image is computed in a traditional manner:

$$NDVI_{\text{image}} = (R_{\text{NIR}} - R_{\text{Red}})/(R_{\text{NIR}} + R_{\text{Red}}) \tag{20}$$

where $R_b$ denotes the reflectance of the image at wavelength $b$ (red or NIR band).

Two kinds of LiDAR NDVIs are computed based on the LiDAR waveforms in the NIR and red bands. The NDVI profile [34] is calculated as:

$$NDVI_i = (B_{i,\text{NIR}} - B_{i,\text{Red}})/(B_{i,\text{NIR}} + B_{i,\text{Red}}), \tag{21}$$

where $B_{i,b}$ is the energy of the waveform at wavelength $b$ and bin $i$.

The NDVI derived from the integrated waveform energy [16] is defined as:

$$NDVI_{\text{integrated}} = \frac{\sum_i B_{i,\text{NIR}} - \sum_i B_{i,\text{Red}}}{\sum_i B_{i,\text{NIR}} + \sum_i B_{i,\text{Red}}} \tag{22}$$

Two waveforms in the red and the NIR bands, respectively, corresponding to a certain emitted pulse are shown in Figure 12. The values of the NIR waveform tend to be larger than the values of the red waveform due to the higher spectral reflectance characteristics of vegetation in the NIR band. An NDVI profile that can be retrieved from multispectral LiDAR waveforms in the NIR and red bands based on Equation (21). The NDVI profile is

related to vertical parameters of vegetation, such as layered LAI and chlorophyll content. It shows that multispectral LiDAR has the capability to obtain more detailed vertical information. The NDVI profile can help to discriminate the contribution from upper or lower vegetation. For example, the reflectance of the understory is similar with that of the overstory so that it is difficult to separate trees from the background in a single reflectance image. However, it is clear from the multispectral LiDAR results that there is a background and at least two separated layers in the plot, which can be seen from the peaks of the LiDAR waveforms correlated with the horizontal levels of the elements. In fact, there are mainly two kinds of trees in the scene. The heights of *Betula pendula* vary from 19.86 m to 30.51 m, and those of *Tilio cordata* vary from 11.27 m to 20.70 m. Thus, *Tilio cordata* crowns are beneath *Betula pendula* crowns. The application of multispectral LiDAR is out of the scope for the purposes of this article so we do not deeply discuss the characteristics of the NDVI profile here.

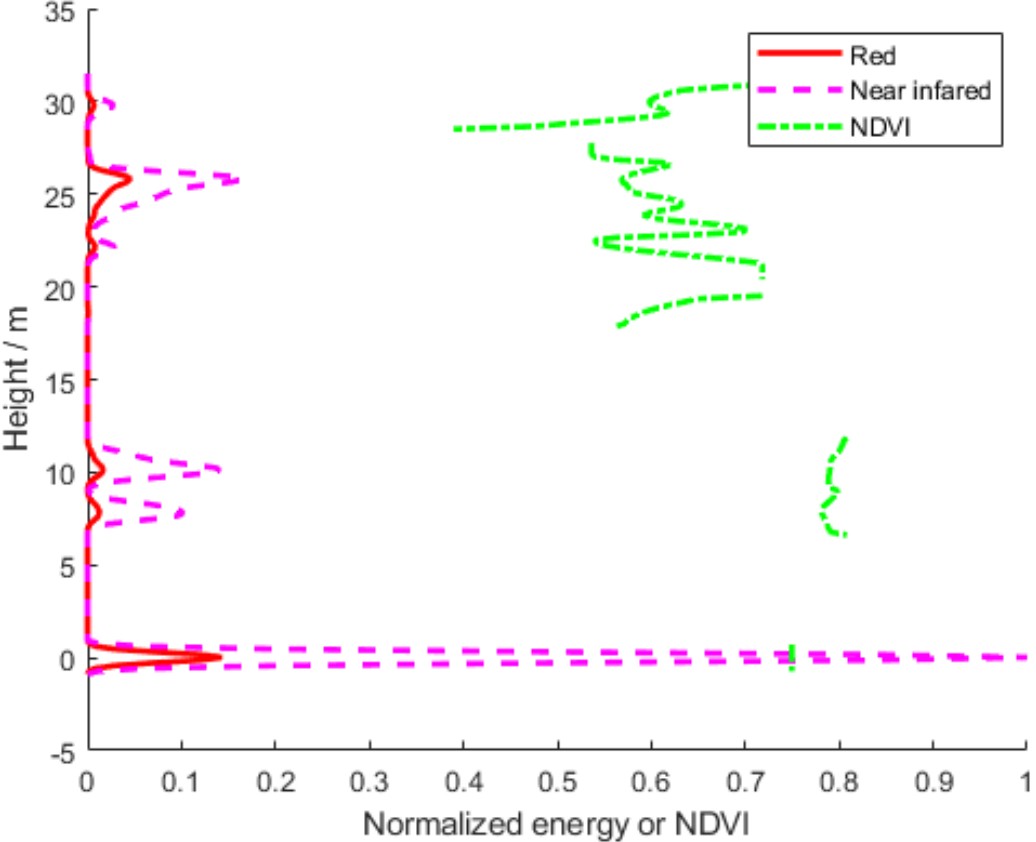

**Figure 12.** An example of multispectral waveforms and the normalized difference vegetation index (NDVI) profile. The NDVI of the background is close to that of the foliage. Therefore, it is difficult to directly separate trees and background in the NDVI image. However, it is clear from the multispectral LiDAR result that there are trees (upper) and a background (lower) with a similar NDVI.

The NDVI values derived from the integrated waveform energy (Equation (22)) are compared with the NDVI values derived from the simulated multispectral images with different incident solar directions (Figure 13). The differences between NDVIs from the integrated waveform energy and the image arise from two features of the illumination conditions. One is related to the geometry of the source–object–sensor. It is known that as anisotropic objects, reflectance from land surface is different with different incident and viewing directions. As an active sensor, LiDAR always receives backward signals (hot spot directions with small zenith angles), while passive sensors often receive signals reflected by the land surface of solar light with different incident and viewing directions. Therefore, it can be seen that there is a better correlation between NDVIs from waveforms

and the image with SZA = 0° (Figure 13a) than those cases with SZA = 30° (Figure 13c) and SZA = 60° (Figure 13e). Another reason is the issue of the adjacency effect. When generating LiDAR data, most of the accumulated contributions come directly from the elements within the footprints, while the whole scene is illuminated by the sun when generating images, and there are contributions of scattering from nearby pixels. There is more vegetation around the pixel, which causes a higher reflectance in the NIR band due to multiple scatterings. Therefore, the NDVI values derived from the images are apparently higher than those from the LiDAR waveform due to the adjacency effect. If considering the first-order scattering results only, the differences of NDVIs from the image and LiDAR decrease obviously and the corresponding coefficients of determination increase (Figure 13b,d,f).

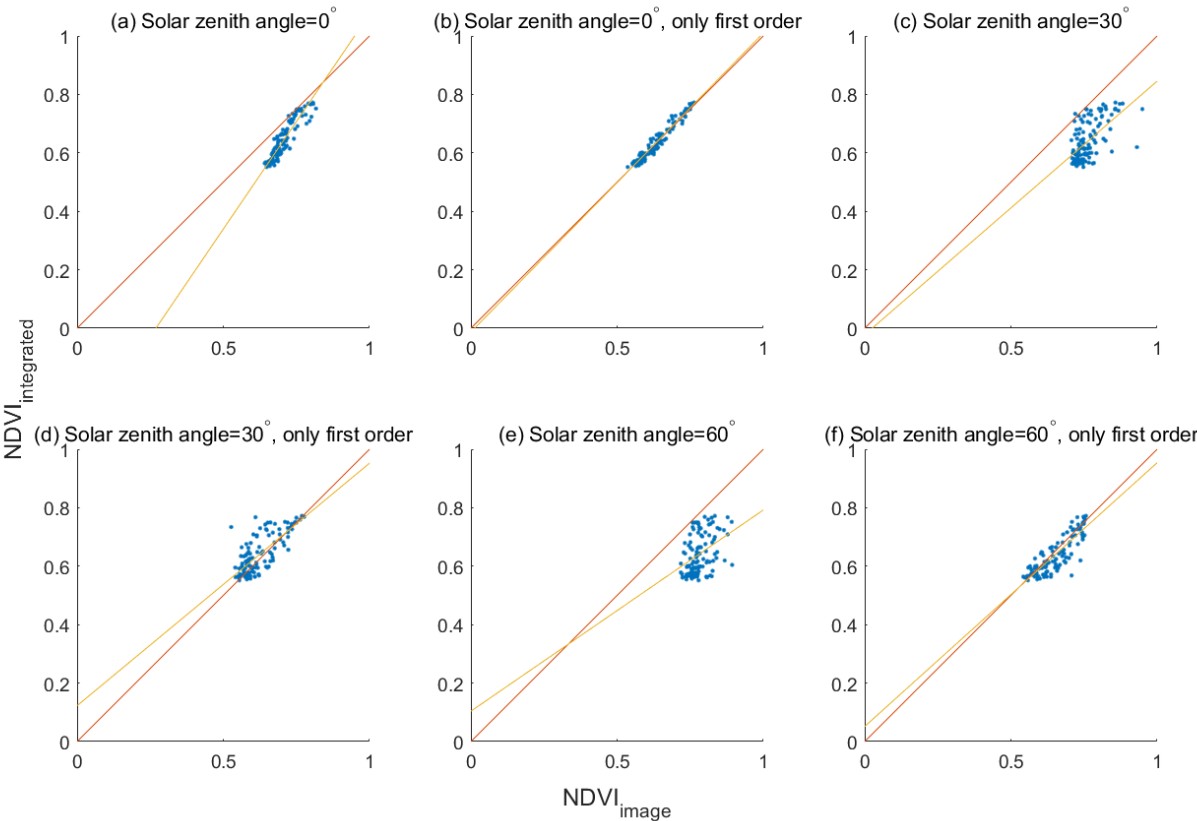

**Figure 13.** Normalized difference vegetation indexes derived from integrated waveform energy ($NDVI_{\text{integrated}}$) versus image ($NDVI_{\text{image}}$). $NDVI_{\text{image}}$ with different solar zenith angles (SZA) in each row and with multiple (left column) and single (right column) scattering (only first order) are calculated and compared with $NDVI_{\text{integrated}}$, separately. SZAs, RMSEs, and $R^2$ values are (**a**) SZA = 0°, RMSE = 0.016, $R^2$ = 0.86, (**b**) SZA = 0°, RMSE = 0.013, $R^2$ = 0.96, (**c**) SZA = 30°, RMSE = 0.038, $R^2$ = 0.36, (**d**) SZA = 30°, RMSE = 0.041, $R^2$ = 0.58, (**e**) SZA = 60°, RMSE = 0.035, $R^2$ = 0.15, and (**f**) SZA = 60°, RMSE = 0.032, $R^2$ = 0.76.

This result shows that the proposed LiDAR simulator fulfills the design goal of simulating multispectral LiDAR signals. It is hoped that LESS and one of its extensions, the LiDAR simulator, which offers a comprehensive simulation environment combining active and passive remote sensing simulations, can help to make multisensor fusion simulations investigations efficient.

*5.2. Performance*

The performance of the LESS LiDAR simulator is evaluated with the DART LiDAR module [17,18] and HELIOS++ [20], which are the most popular LiDAR simulators.

In order to test the performance efficiency, the scene built from the RAMI-V data (Section 2.2) is used. Point cloud simulations are carried out on an Intel(R) Xeon(R) Silver 4110 CPU @ 2.10 GHz with 64.0 GB of random access memory (RAM). The angular sampling for both zenith and azimuth angle is set as $0.06°$. The azimuth range is $0°$–$360°$, and the zenith range is $30°$–$130°$. Therefore, 10 million laser pulses will be sent with different directions.

The time consumption and the output point numbers are given in Table 8. Note that the LESS LiDAR simulator requires an additional configuration time to prepare a configuration file about the situations of the moving platform, including LiDAR locations and sending directions, which depend on the number of sending directions. In this case, it is about 4 min.

**Table 8.** Performance comparison of DART LiDAR, HELIOS++, and LESS LiDAR.

|  | DART LiDAR | HELIOS ++ | LESS LiDAR |
|---|---|---|---|
| Configuration time/min | - | - | 4 |
| Loading scene time/min | 6 | 37 | $4 \times$ (2 times) |
| Simulation time/min | 135 | 100 | 28 |
| Total time/min | 141 | 137 | 40 |
| Point number/$10^6$ | 6.4 | 7.6 | 6.6 |
| Ground point number/$10^6$ | 3.6 | 3.3 | 3.6 |
| Non-ground point number/$10^6$ | 2.8 | 4.3 | 3.0 |

The loading scene time and simulation time are given for these three models, separately. The loading scene time "$4 \times$ (2 times)" for LESS LiDAR means the scene is loaded 2 times, and each time takes around 4 min. Because the issue of running on computers with smaller memory is considered, the simulation is simply split into several parts in this version. This method helps LESS LiDAR to run on a computer low on RAM.

In addition, LESS uses an "instance" technique, which only keeps one copy of each object in memory, and stores the geometric transforms of all individual trees. This technique can save more memory making it good for simulating a bigger scene with many similar or same objects.

The simulation time also depends on the number of rays in each laser beam. Both the DART LiDAR module and LESS LiDAR simulator use the axial division of 10, causing $10^2 \times \pi/4 \approx 79$ rays for each beam. HELIOS++ uses a parameter called the beamSampleQuality to control the number of rays for each beam. The beamSampleQuality corresponds to the number of concentric circles where rays are sampled. If the beamSampleQuality is $s$, the number of rays in each beam is $1 + \sum_{i=1}^{s} \lfloor 2\pi i \rfloor$. This ensures that the angular distance between adjacent rays is approximately constant, i.e., each ray represents a solid angle of equal size [20]. If the beam sample quality is 5, there are 61 rays for a beam. And if the beam sample quality is 6, there are 98 rays for a beam. Both of them are close to the 79 used in the DART module and LESS LiDAR simulator. To simulate efficiently, 61 rays for a beam is chosen for HELIOS++. More rays makes little difference to the results.

The total time is the sum of the configuration time, loading scene time, and simulation time. It it obvious that the LESS LiDAR simulator requires the shortest time among the three models, and its simulation efficiency is much higher than the DART LiDAR module and HELIOS++. It is important to note that we performed simulations using identical input files across all models. However, we acknowledge that this comparison may have introduced some biases because the input procedures did not optimize or leverage the unique design aspects of each model. Consequently, there is potential for further improvements in efficiency by fine-tuning the data loading settings specific to each model.

The total returned point numbers are also listed in Table 8, and it can be seen that there are a few differences among the three models. LESS LiDAR obtains 6.6 million points in this case, which is between DART LiDAR and HELIOS++. LESS LiDAR and DART LiDAR achieve similar ground points. HELIOS++ has the most number of the total points, while the least number of ground points. DART LiDAR has the least non-ground points. All the sets of simulated points can be registered well with each other and the differences perhaps relate to the parameters and data processing methods used in each model.

### 5.3. Multi-Ray Point Cloud vs. Single-Ray Point Cloud

The procedure described in Section 3 allows for the simulation of LiDAR waveforms and their corresponding points for a single pulse. By running this procedure in a loop, it is possible to simulate multi-pulse data.

As mentioned in Section 3, laser beams are not infinitely thin lines, but rather cones of light that intersect surfaces within an elliptical area known as the laser's footprint. This characteristic has implications for how light is reflected by surfaces and detected by LiDAR sensors. Modern laser scanners take these effects into account and measure the full waveform of a reflected laser pulse. The full waveform contains valuable additional information that can be used by LiDAR researchers to gain insights into scanned objects. To simulate the divergence of a laser beam, multiple ray casting queries are used to approximate the cone. This mode is referred to as the multi-ray point cloud simulation.

In addition to the multi-ray approach, another simulation method known as the single-ray point cloud simulation has been developed. This method is efficient when simulating point clouds from targets located at a short distance with a small divergence angle of the laser beam. In such cases, where the footprint is very small and the beam can be treated as one ray, each point's position is determined by the coordinates of its first intersection with a surface. This method is efficient due to reducing the ray casting queries.

To evaluate and compare the performance between the single-ray point cloud simulation mode and multi-ray point cloud simulation mode, Table 9 presents relevant metrics.

**Table 9.** Performance comparison of single-ray point cloud and multi-ray point cloud.

|  | Single-Ray Point Cloud | Multi-Ray Point Cloud |
|---|---|---|
| Simulation time/min | less than 1 | 28 |
| Point number/$10^6$ | 6.2 | 6.6 |
| Ground point number/$10^6$ | 3.5 | 3.6 |
| Non-ground point number/$10^6$ | 2.7 | 3.0 |

With respect to simulation time, it was found that the single-ray point cloud mode demonstrated a significant advantage over its multi-ray counterpart, completing simulations in less than 1 min compared to the 28 min required by the latter.

Considering the numbers of points generated by both modes, similar values were observed: approximately 6.2 million points for the single-ray mode and 6.6 million points for the multi-ray mode. However, it is noteworthy that the single-ray point cloud mode exhibited slightly fewer points, approximately 6.0% less than the multi-ray point cloud mode.

Further analysis reveals comparable numbers of ground points in both modes. The single-ray point cloud mode yielded 3.5 million ground points, while the multi-ray point cloud mode generated 3.6 million ground points, resulting in a difference of only 2.7%.

In terms of non-ground points, notable discrepancies emerged between the two modes. The single-ray point cloud mode produced approximately 2.7 million non-ground points, whereas the multi-ray point cloud mode generated slightly more at 3.0 million non-ground points, representing an approximate reduction in non-ground point numbers by about 10% when using the single-ray simulation approach compared to employing multiple rays.

It should be emphasized that for scenes with flattening ground surfaces, such as observed here, there is typically just one generated ground point per location; hence, there is no substantial impact on ground points.

It is important to note that non-ground points mainly originate from vegetation containing scattered leaves and branches, which introduce greater complexity into how light is reflected by surfaces and detected by LiDAR sensors. In this regard, there are often multiple intersections along a pulse beam path that can be captured by the multi-ray mode but are missed or only roughly approximated by the single-ray point simulations since they rely on recording only the coordinates of the first intersection with a surface. Consequently, using single-ray simulations may result in more points being detected by LiDAR sensors

and an approximate reduction in non-ground point numbers by about 10% compared to using multiple rays.

Taken together, these findings demonstrate that utilizing a single-ray simulation approach can significantly reduce simulation time without substantially affecting overall point numbers or the ground pointswhen compared to employing multiple rays for simulations.

In summary, the single-ray point cloud simulation mode proves to be a valuable option for scenarios where relatively short distances and small footprints of the laser are involved (such as when flat ground is large enough compared to the footprints in this simulation). It allows for efficient simulations and its reduced simulation time makes it a practical choice. However, researchers should consider the potential loss of detailed information in fragmented objects when using this mode. On the other hand, the multi-ray mode simulates the divergence of a laser beam to generate simulated full waveforms containing valuable additional information that can be used by LiDAR researchers to gain insights into scanned objects.

## 6. Conclusions

A LiDAR simulator based on the ray tracing algorithm was developed, which is suitable for the complex landscapes of vegetation canopies with high horizontal and vertical heterogeneity. The new LiDAR simulator can simulate full waveform and point cloud signals from LiDAR loaded on different platforms, including airborne, unmanned aerial vehicles, and terrestrial LiDAR. Furthermore, it is inferred that space-borne LiDAR signals can also be simulated using LESS LiDAR as they adhere to the same underlying principles, although some issues should be considered, such as the atmospheric attenuation and the solar noise as a light source.

The proposed LiDAR simulator has the capability to simulate multispectral LiDAR data. Multispectral waveforms were simulated and NDVIs derived from waveforms and images were compared to simply analyze the characteristics of multispectral LiDAR. Realistically reconstructed landscapes can be used in the LiDAR simulator to adapt the comparable actual data. Although this is beyond the scope of this study, a more refined 3D model including various disturbances and optical characteristics would be needed to better evaluate the effects of instrument design and laser interactions with different surface features.

By considering both the accuracy and performance results together, it becomes apparent that the proposed LiDAR simulator enables efficient simulations without sacrificing accuracy.

In summary, our novel LiDAR simulator holds immense potential in advancing research within remote sensing studies and related fields. With its comprehensive simulation capabilities and improved efficiency compared to other models, it serves as an invaluable tool for the remote sensing community. Notably, several applications have already been showcased. For instance, the LESS model was employed to generate simulated data for unmanned aerial vehicle (UAV) LiDAR, enabling exploration of the potential for trunk point extraction and direct diameter-at-breast-height measurements through UAV LiDAR [35]. Furthermore, the potential of airborne hyperspectral LiDAR in assessing forest insects and diseases has been evaluated using LiDAR simulations [36].

With regards to the limitations of the study, several ideal assumptions were made. The future extensions of the proposed LiDAR simulator will consider solar noise, atmospheric attenuation, as well as the response function, gain, and noise of the LiDAR system. As for the performance improvements, LiDAR simulations can have high computational cost, which demands efficient computational solutions. It is suggested that LiDAR simulators may benefit from high-performance computing (HPC) algorithms or techniques. For instance, using HELIOS++ as a case study, novel algorithms run on parallel computers boost the performance [37]. It is worth further investigating how the LESS LiDAR simulator can benefit from these algorithms or techniques.

The proposed LiDAR simulator was integrated into LESS, where an easy-to-use graphical user interface and related tools for configuring the simulations were provided. The purpose of extending the LESS model is to simulate passive images and active LiDAR signals based on the same configurations and parameters. The software of the integrated LESS can be downloaded from https://lessrt.org (accessed on 10 September 2023), providing users with a binary installer. Individuals can refer to the documentation for instructions on scene configuration and parameter adjustment.

**Author Contributions:** Funding acquisition, D.X.; software, Y.L. and J.Q.; writing—original draft, Y.L.; writing—review and editing, D.X. and J.Q.; investigation, K.Z.; supervision, G.Y. and X.M. All authors have read and agreed to the published version of the manuscript.

**Funding:** The work is funded by the National Natural Science Foundation of China (Grant No. 42071304 and 42090013) and the National Key Research and Development Program of China (Grant No. 2020YFA0608701 and 2022YFB3903304).

**Data Availability Statement:** The data presented in this study are available on request from the corresponding author.

**Acknowledgments:** The authors thank Zhonghu Jiao, Hailan Jiang, and Yingjie Wang for giving nice suggestions.

**Conflicts of Interest:** The authors declare no conflict of interest.

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
