# Peer review of "LESS LiDAR: A Full-Waveform and Discrete-Return Multispectral LiDAR Simulator Based on Ray Tracing Algorithm"

_remotesensing, doi:10.3390/rs15184529_

Round 1
Reviewer 1 Report
Comments
1) The abstract need to add the significant quantitative findings.
2) What are the gaps from the previous study?
3) What is the novelty of this study? The author should highlight it in a perfect academic writing style with thorough explanations.
4) Please elaborate on the findings and add more to the cross-validation statement to support your study.
5) The conclusion should be comprehensive, including the findings from the study, the significant results and the impact.
Reviewer 2 Report
General comments:
This paper proposed a full-waveform and discrete-return multispectral LiDAR simulator called LESS-LiDAR based on the ray tracing algorithm. The performance of LESS-LiDAR simulator was evaluated by comparing it with the widely-used DART and HELIOS.
LiDAR simulator is important for sensor design and the development of remote sensing algorithms. I believe it is an important study that will provide an important and efficient LiDAR simulation tool for the remote sensing community. I would recommend a minor revision before accepting it for publishing.
Specific suggestions:
1. L12: I would use “point clouds” instead of points.
2. L50: “the limitation of 3D scene scale” is confusing.
3. L118-121: a citation is necessary.
4. L141: I suggest revising “LiDAR returns” to “discrete returns”.
5. L53: The laser beam has a divergence, which has no relation with the length of the beam path.
6.L210:Reviese “estimate” to estimation.
7. L483: In what circumstances would the single ray point cloud simulation be used? I suggest a discussion about it.
8. L494-495: According to the manuscript, I think space-borne LiDAR signal can also be simulated from LESS-LiDAR.
9. L101: I would use “Realistic” instead of “Actual”.
10. Figure 3: whether it is reflectance or transmittance should be clarified.
11. L163: beam.
12. The title of the manuscript:
I suggest deleting “multispectral” in the title. According to the manuscript and my personal experience in using other LiDAR simulators, my understanding is that hyperspectral or single-band LiDAR signals can also be simulated via LESS-LiDAR. Anyway, I leave this to the authors.
Reviewer 3 Report
1. The citations of introduction should be more focused on the paper's topic, some of citations can not be used to support author's point.
2. Some of abbreviation in the figure shoud be give full name to increase article readability.
3. The innovation points of paper need to be further clarified.
Some of abbreviation in the figure shoud be give full name to increase article readability.
